# Periodontal Disease in Patients Receiving Dialysis

**DOI:** 10.3390/ijms20153805

**Published:** 2019-08-03

**Authors:** Yasuyoshi Miyata, Yoko Obata, Yasushi Mochizuki, Mineaki Kitamura, Kensuke Mitsunari, Tomohiro Matsuo, Kojiro Ohba, Hiroshi Mukae, Tomoya Nishino, Atsutoshi Yoshimura, Hideki Sakai

**Affiliations:** 1Department of Urology, Nagasaki University Graduate School of Biomedical Sciences, Nagasaki 852-8501, Japan; 2Department of Nephrology, Nagasaki University Hospital, Nagasaki 852-8501, Japan; 3Division of Blood Purification, Nagasaki University Hospital, Nagasaki 852-8501, Japan; 4Department of Respiratory Medicine, Unit of Basic Medical Sciences, Nagasaki University Graduate School of Biomedical Sciences, Nagasaki 852-8591, Japan; 5Department of Periodontology and Endodontology, Nagasaki University Graduate School of Biomedical Sciences, Nagasaki 852-8501, Japan

**Keywords:** periodontal disease, peritoneal dialysis, hemodialysis, immune response, diabetes

## Abstract

Chronic kidney disease (CKD) is characterized by kidney damage with proteinuria, hematuria, and progressive loss of kidney function. The final stage of CKD is known as end-stage renal disease, which usually indicates that approximately 90% of normal renal function is lost, and necessitates renal replacement therapy for survival. The most widespread renal replacement therapy is dialysis, which includes peritoneal dialysis (PD) and hemodialysis (HD). However, despite the development of novel medical instruments and agents, both dialysis procedures have complications and disadvantages, such as cardiovascular disease due to excessive blood fluid and infections caused by impaired immunity. Periodontal disease is chronic inflammation induced by various pathogens and its frequency and severity in patients undergoing dialysis are higher compared to those in healthy individuals. Therefore, several investigators have paid special attention to the impact of periodontal disease on inflammation-, nutrient-, and bone metabolism-related markers; the immune system; and complications in patients undergoing dialysis. Furthermore, the influence of diabetes on the prevalence and severity of manifestations of periodontal disease, and the properties of saliva in HD patients with periodontitis have been reported. Conversely, there are few reviews discussing periodontal disease in patients with dialysis. In this review, we discuss the available studies and review the pathological roles and clinical significance of periodontal disease in patients receiving PD or HD. In addition, this review underlines the importance of oral health and adequate periodontal treatment to maintain quality of life and prolong survival in these patients.

## 1. Introduction

Chronic kidney disease (CKD) is defined as a specific, irreversible loss of functional nephrons characterized by progression towards end-stage renal disease (ESRD). The loss of renal function is the most severe form of CKD. In general, when renal function decreases below approximately 10% of normal efficiency, renal replacement therapy is necessary to maintain survival.

Dialytic therapy, mainly peritoneal dialysis (PD) and hemodialysis (HD), are common renal replacement therapies that are used worldwide. Both techniques are performed to remove excessive fluids, electrolytes, and uremic toxins. PD uses the peritoneum as the membrane through which fluids and substances are exchanged with the blood. Solute clearance occurs by solute diffusion from the plasma into a dialysate or ultrafiltration is driven by the osmotic gradient between the hyperosmotic dialysate and the plasma. In contrast, in HD, the transfer between blood and dialysis fluid is performed using a dialyzer membrane. Furthermore, PD has been reported to have the advantage of maintaining residual renal function (RRF) and achieving better outcomes than HD during the first few years of treatment [1]. Other benefits include greater effectiveness in improving quality of life (QoL) and better tolerability in patients with decreased cardiac function. However, PD is less efficient at removing wastes from the body than HD, and the presence of the catheter presents a risk of peritonitis due to the possibility of microbial entry into the abdomen.

Periodontal disease is an oral, chronic infectious, and inflammatory disease caused predominantly by gram-negative anaerobic bacteria, and is characterized by the destruction of tooth-supporting tissues, including the alveolar bone and connective tissues of the periodontium [2,3]. Currently, there is a general agreement that the prevalence of periodontal disease in patients undergoing dialysis is higher than that in healthy individuals. In fact, Borawski et al. [4] reported a marked increase in periodontitis in CKD patients, including patients undergoing predialysis, PD, and HD, compared with the general population. Conversely, when the prevalence and severity of periodontal disease are stratified according to PD or HD treatment, the observed rate was as high as 42.6% in continuous ambulatory PD (CAPD) patients [5]. Moreover, Cengiz et al. [6] reported that the prevalence of moderate to severe periodontitis was 67.3% in CAPD patients. However, in contrast to these results, there have been several reports indicating that PD patients and healthy individuals show a similar prevalence of periodontitis [7,8]. Surprisingly, a recent report suggested that 106 of 107 HD patients (99.1%) exhibit some form of periodontitis [9], and another study also showed that only one of 103 HD patients evaluated had a healthy periodontium [10]. Even if such reports are excluded, many studies have reported that over half of HD patients exhibit periodontitis [11,12,13,14]. Furthermore, most periodontal parameters in HD patients were reported to be significantly higher than those in age-matched control subjects and healthy individuals [15,16,17]. Thus, many investigators have shown that periodontal disease is an important issue in patients with PD and HD.

In this review, we searched for the literature on “periodontal disease” or “periodontitis” and “peritoneal dialysis” or “hemodialysis” using PubMed. We subsequently excluded studies without clinical and laboratory data, and prioritized the most recent studies and those with comparative analyses. We also discuss periodontal indices, the biochemical profile of the blood, and the molecular mechanisms involved in periodontal disease in patients with PD. We focus on the impact of periodontal disease on pathological mechanisms including inflammation, the immune response, and bone metabolism in HD patients. In addition, we compare the severity of periodontal disease, periodontal parameters, and oral health-related conditions in HD patients with diabetic and non-diabetic nephropathies.

## 2. Peritoneal Dialysis

### 2.1. Impact of Periodontal Disease

In this review, different measures of the severity of periodontal disease are discussed. These measures and their criteria are familiar essentially to dentists. Therefore, we present a summary of some representative measures Table 1 [4].

There have been several studies to date indicating that longer duration of CAPD is associated with the severity of periodontitis [5,6]. In addition, the severity of periodontitis correlated positively with levels of inflammatory parameters (high-sensitivity C-reactive protein (hs-CRP), serum ferritin, and white blood cell count) and atherosclerotic risk factors (serum low-density lipoprotein cholesterol, lipoprotein (a), and homocysteine) [5,6]. Conversely, periodontal health status showed a significant negative correlation with serum albumin and blood urea nitrogen (BUN) levels, which suggest poor nutritional status [6]. In contrast to HD, the timing of blood sampling did not have a significant impact in CAPD. Therefore, we believe that periodontal conditions affect the inflammatory and nutritional parameters in PD patients’ blood samples.

Hepatocyte growth factor (HGF) is known to play important roles in embryogenesis, morphogenesis, wound repair, and tissue regeneration [18]. Oshima et al. reported that these pleiotropic properties of HGF might be involved in the development and progression of periodontal diseases [19,20,21]. Previously, Wilczynska-Borawska et al. [22] showed that there was no difference in HGF levels in the saliva of HD, PD, and chronic renal failure patients, although the levels in the saliva of periodontitis patients were higher than those in the healthy population. In PD patients specifically, significant and positive correlations between HGF levels in the saliva and plaque index (PI), papillary bleeding index (PBI), and gingival index (GI) have been reported. Similar to blood sampling, the timing of saliva collection may not be a factor influencing HGF levels in patients on PD. Thus, HGF might contribute to the development of periodontal disease in PD patients. However, the molecular mechanism involved in the pathogenesis of periodontal disease in PD patients remains to be clarified. Herein, we hoped to clarify the precise mechanisms of pathogenesis and progression of periodontal diseases.

### 2.2. Impact of Periodontal Care and Treatment

Several studies have investigated the effects of periodontal therapy in PD patients [5,22]. Briefly, treatment of periodontal disease improved periodontal status, inflammatory markers, and nutrition status [5]. Tasdemir et al. [23] investigated the effects of periodontal therapy on inflammation markers in PD patients with diabetic nephropathy, diabetic patients without CKD, and in the healthy population, since it has been reported that diabetes mellitus (DM) is one of the risk factors of periodontitis, and that periodontal inflammation causes poor glycemic control. The authors demonstrated that all inflammatory markers including tumor necrosis factor (TNF)-α, pentraxin-3 (PTX-3), interleukin (IL)-6, and hs-CRP in blood samples were significantly higher in PD patients with diabetic nephropathy than in the other two groups, and TNF-α was reduced after 3 months of periodontal treatment in all patients [23]. Conversely, PTX-3, IL-6, and hs-CRP levels were decreased after periodontal treatment only in PD patients with diabetic nephropathy [23]. Thus, the authors speculated that periodontal disease is a major source of inflammation in CAPD patients with diabetes [23]. Conversely, in a study evaluating the clinical impact of PD on oral health, Keles et al. [23] reported that the degree of halitosis was significantly reduced by PD therapy. As a potential underlying mechanism, the authors speculated that a decrease in BUN levels and an increase in salivary flow rates (SFR) resulting from adequate PD treatment might be associated with the improvement of halitosis, as high BUN levels and low SFRs have been reported to play important roles in the severity of halitosis [24]. Thus, periodontal care and treatment are useful for ameliorating a variety of the inflammatory manifestations in diabetic nephropathy patients, and for partially relieving symptoms caused by periodontal diseases. However, further studies in non-diabetic patients, with a focus on periodontal disease-related symptoms, are necessary before drawing definitive conclusions.

## 3. Hemodialysis

### 3.1. Correlations with Blood Test Indices

As mentioned above, HD is a form of renal replacement therapy, whereby toxins, waste, and excessive proteins and electrolytes, such as urea, potassium, and excess fluids are removed from the blood. This enables HD to improve QoL and prolong survival in patients with ESRD. Unfortunately, bacterial infection is common in patients with HD, and is a major cause of death because of the suppression of immunological function, increasing incidence of diabetes, and deterioration of nutritional status [25,26,27]. In HD patients, periodontal disease induces not only local inflammation but also systematic inflammatory responses [28,29,30]. In a QoL analysis of HD patients, 5 of 8 indices of the SF-36 health scale (physical functioning, role physical, vitality, social functioning, and mental health) were significantly lower in patients with severe periodontitis (*p* < 0.05) compared to those with no/mild periodontitis [14]. Thus, many studies have investigated the relationships between the prevalence and/or severity of periodontal disease and serum levels of inflammation-related factors in HD patients. Moreover, periodontal disease may also affect nutritional and bone loss parameters in HD patients [10,31]. Based on these reports, several researchers and clinicians have paid special attention to the relationship between periodontal disease and changes in systemic conditions, including inflammation, nutrition, and bone metabolism, resulting from HD. Therefore, we have reviewed the data on serum levels of CRP, albumin, calcium, phosphorus, parathyroid hormone (PTH), and other hematological parameters in HD patients. However, we should note the fact that the timing of blood sampling is not clearly defined in almost all of the previous reports. In general, routine blood sampling is performed at the initiation of HD. However, if it is performed at the time of a dental consultation, sampling at the initiation of HD is not possible. In real-world settings, the overall management of patients and supervision of HD is performed by nephrologists or urologists. Therefore, blood sampling can be assumed to have been performed at the initiation of HD.

#### 3.1.1. Serum CRP Levels

Serum CRP levels in HD patients with advanced periodontitis have been reported to be significantly higher (*p* < 0.05) than in those without periodontitis [32]. Furthermore, other investigators have shown that serum CRP levels were decreased following periodontal therapy (*p* = 0.001) in HD patients with periodontal disease [30]. Moreover, the number of teeth was negatively associated with serum CRP level in patients with HD (*r* = −0.23, *p* < 0.05) [10]. These results support the notion that serum CRP levels may reflect the inflammatory status of periodontal tissues, and that the serum CRP level is a useful marker of treatment success in HD patients with periodontal disease. In contrast to these findings, in 154 patients on HD, the mean serum CRP level in those with severe periodontitis was reported to be similar to that in those without periodontitis (9.27 and 11.90 mg/dL, respectively; *p* = 0.28) [33]. In addition, the same study found no significant difference in serum CRP levels (*p* = 0.23) between HD patients with no/mild periodontitis (*n* = 100) and moderate/severe periodontitis (*n* = 68). This non-significant relationship between serum CRP levels and severity of periodontal disease was supported by other investigators [10,12]. Moreover, salivary CRP levels in HD patients were reported to be significantly higher compared to both controls and patients with CKD not receiving HD [34]. Thus, there are contrasting opinions regarding the pathological significance of serum CRP levels in HD patients with periodontal disease.

Conversely, a study measuring hs-CRP levels showed that the median (interquartile range) serum level in HD patients with no, mild, moderate, and severe periodontitis was 1.96 (0.79–8.17), 2.72 (0.87–6.91), 4.19 (1.92–10.47), and 4.42 (2.46–13.4), respectively, showing a significant positive correlation (*p* = 0.008) [35]. Likewise, serum hs-CRP levels have been reported to be a significant and independent predictor for the development of periodontal disease in a multivariate model that included DM, frequency of teeth brushing, and various serum markers in HD patients [36]. Based on these results, we felt that hs-CRP was a better marker of the severity of periodontal disease and progression of the disease in HD patients than CRP.

Regarding changes in serum CRP level following treatment of periodontal disease in HD patients, conflicting results have been reported. For example, one report showed that serum CRP levels were significantly decreased (*p* = 0.001) after periodontal treatment in 41 patients receiving HD [30]. Similar results have also been reported in 77 HD patients treated with non-surgical methods [12]. In contrast, other investigators have reported that serum CRP levels in HD patients with treated chronic periodontitis (*n* = 43) were similar (*p* = 0.634) to untreated patients (*n* = 30) [37]. Conversely, a study measuring hs-CRP showed that serum levels were significantly decreased (*p* < 0.001) by the treatment of periodontal diseases, including non-surgical and surgical methods (mean/SD: 3.8/21.9 to 0.6/5.9 mg/L) [38]. As mentioned above, there is the possibility that hs-CRP levels are a better indicator of the inflammatory status caused by periodontal disease in HD patients. However, it should be noted that not only periodontal diseases but also other factors are recognizable as sources of inflammation in patients with HD [39,40]. Further studies with more detailed analyses and larger populations are necessary to determine the pathological significance and value of CRP or hs-CRP levels as biomarkers in HD patients with periodontal disease. A summary of these results is shown in Table 2.

#### 3.1.2. Serum Albumin Levels

Evaluation of nutritional status in HD patients is important because hypo-albuminemia is known as a predictive factor of worse mobility and mortality [42,43]. Many investigators have paid special attention to the relationship between periodontal disease and serum albumin levels. Kshirsagar et al. reported that the mean/SD serum albumin levels in HD patients with and without periodontitis (*n* = 35 and 119, respectively) were 3.83/0.41 and 3.99/0.53 mg/dL, respectively, although this difference did not reach statistical significance (*p* = 0.06) [33]. However, their study also showed that severe periodontitis was significantly associated with low serum albumin levels in univariate logistic regression analysis (odds ratio = 3.23, 95% confidential interval [CI]; 1.16–8.96, *p* = 0.02) [33]. In addition, the same significant correlation was confirmed by multivariate analysis models including clinical features and laboratory data [33]. A positive association between periodontitis and hypo-albuminemia was confirmed by other investigators [13]. Furthermore, several reports have demonstrated that serum albumin levels were negatively associated with representative parameters of periodontal health status including plaque index (*r* = −0.26, *p* < 0.01), (*r* = −0.28, *p* < 0.01), periodontal disease index (*r* = −0.29, *p* < 0.01), and pocket depth (*r* = −0.20, *p* < 0.05) [10,35], and a multivariate analysis showed that the serum albumin level is an independent predictor of the periodontal disease index (relative ratio = −0.47, CI = −0.91 to 0.03, *p* = 0.036) [35]. Moreover, serum albumin levels in HD patients with treated chronic periodontitis were significantly lower (*p* = 0.023) than in untreated patients [37]. Thus, periodontal disease seems to decrease serum albumin level in patients with HD. This negative correlation can be explained by various reasons, including protein energy malnutrition, consistent inflammation, and reduced oral intake [33]. However, in patients with HD, there is no general agreement on this relationship, and the detailed mechanism is not fully understood. In fact, a recent report indicated that serum albumin levels were not significantly different between dentate HD patients and edentulous patients (*p* = 0.761), or patients with healthy periodontium or gingivitis and those with periodontitis (*p* = 0.601) [10].

In contrast to these findings, HD patients with moderate-severe periodontitis exhibited higher serum albumin levels (mean/SD; none or mild periodontitis: 3.7/0.4 g/dL and moderate or severe periodontitis: 3.9/0.4 g/dL) [44]. In addition, another report suggested that serum albumin levels increased following treatment of periodontal diseases (mean/SD: 3.15/0.30 to 3.38/0.37 g/dL) in 30 patients with HD [38]. We have no clear answer for these differing findings. However, besides nutritional status, serum albumin levels are regulated by various pathological conditions including aortic calcification, peptic ulcer diseases, and body fat mass [45,46,47]. Furthermore, we should note the method of statistical analysis used for each study. Briefly, one study showed that the frequency of patients with hypo-albuminemia (<3.6 g/dL) was not significantly associated with periodontitis in HD patients [14,41], the same group also showed that serum albumin levels in HD patients with periodontitis was significantly lower (*p* = 0.01) compared to patients without periodontitis [41]. This information is shown in Table 3.

#### 3.1.3. Calcium, Phosphorus, Alkaline Phosphatase, and PTH

Secondary hyper-parathyroidism and 1,25-dihydroxy vitamin D3 deficiency are common and important complications in HD patients. Furthermore, they can lead to bone fracture and arthropathy via the reduction in bone density, and changes in serum calcium, and phosphorus, while PTH levels play important roles in pathogenesis and progression of HD-related complications. However, there is a report indicating that alkaline phosphatase is a useful marker for the diagnosis of periodontal disease [49]. Therefore, we will discuss the relationship between periodontal disease and serum levels of calcium, phosphorus, alkaline phosphatase, and PTH in patients with HD.

As shown in Table 3, to our knowledge, all reports that have investigated changes in serum calcium levels by periodontitis in HD patients showed no significant difference [10,13,36,48]. In contrast, a cross-sectional study reported that higher calcium intake (584.5–1478.5 mg/day) was inversely associated with probing pocket depth (PPD) > 4 mm (adjusted odds ratio; 0.53, 95% CI; 0.30–0.94, *p* = 0.03) compared to a lower intake group (230.7–393.4 mg/day) [50]. Based on this result, supplementation with calcium might be useful to prevent periodontal disease in HD patients [36]. However, we must consider the fact that the study population comprised young women with a mean age of 31.5 years and normal renal function [50].

Regarding phosphorus in HD patients, the mean/SD serum levels in healthy/gingivitis patients (5.87/1.59 mg/dL) showed a trend towards being higher than in moderate/severe periodontitis (5.29/1.68 mg/dL) patients; however, this difference did not reach statistical significance (*p* = 0.084) [10]. In addition, several investigators have shown that serum phosphorus levels in periodontal disease are not significantly different compared to non-gingivitis groups [36,48]. However, in contrast to these results, other investigators showed that serum phosphorus levels in HD patients with periodontitis (mean/SD; 5.02/1.19 mg/dL) were significantly lower (*p* = 0.024) than in patients without periodontitis (6.25/1.72 mg/dL) [13]. In addition, serum phosphorus levels have been reported to be positively correlated with clinical attachment loss (CAL; *r* = 0.47, *p* = 0.037) [48]. However, the above study also showed that serum phosphorus levels were not associated with other periodontal parameters, such as PD, PI, GI, or bleeding on probing [48]. Conversely, serum phosphorus levels in patients with untreated chronic periodontitis (6.1/1.2 mg/dL) have been reported to be similar (*p* = 0.221) to those of treated patients (6.5/1.2 mg/dL) [37]. Although further studies are necessary, it appears that serum phosphorus levels might not influence periodontal disease in HD patients based on the calcium and PTH levels observed (see below).

Several studies have reported that periodontal disease had no significant correlation with serum PTH levels in HD patients [10,35,36,51]. In addition, alveolar bone loss was not correlated to serum PTH level in 35 HD patients with secondary hyper-parathyroidism [51]. Finally, the authors speculated that secondary hyper-parathyroidism and increased serum PTH levels played minimal roles in periodontal disease and periodontal indices in HD patients, a speculation that we also find plausible.

Conversely, as shown in Table 3, several reports have shown no significant correlation between serum levels of alkaline phosphatase and periodontal diseases [10,13]. In addition, serum alkaline phosphatase levels in HD patients with untreated chronic periodontitis (mean/SD: 134/145 mg/dL) showed a trend towards higher levels than in treated patients (117/70 mg/dL); however, this difference was not statistically significant (*p* = 0.687) [37]. Thus, there is no evidence that periodontal disease affects serum alkaline phosphatase levels. Unfortunately, there is little information on changes in serum bone-specific alkaline phosphatase levels related to periodontal disease in HD patients. In an animal model, experimental periodontitis was reported to affect serum levels of bone-specific alkaline phosphatase [52]. Thus, there is a possibility that bone-specific alkaline phosphatase reflects periodontal bone loss and/or its metabolism in patients with HD. These results are shown in Table 4.

#### 3.1.4. Hematological Data

Several investigators have reported that there was no significant correlation between periodontitis and hematological data, such as white blood cell and platelet counts, and hematocrit [11,13,28,35,48]. However, others have found that the presence of periodontal disease was positively correlated with white blood cell count (*p* < 0.001), but not with hemoglobin levels or platelet count [36]. Such increases in white blood cell count might be due to local and/or systematic inflammation. However, to our knowledge, a significant correlation between periodontal disease and white blood cell counts was found in only one study [36]. In addition, the number of white blood cells in the gingival crevicular fluid of HD patients with periodontal disease (mean/SD: 6.05/1.81 k/μL) was significantly lower (*p* < 0.001) than that in a control group (7.02/1.30 k/μL) [53]. Based on these findings, the impact of inflammation on circulating white blood cell counts in HD patients with periodontal disease can be considered minimal.

Periodontal treatment has also been reported to significantly increase hemoglobin levels, from 9.4 g/dL to 10.6 g/dL (*p* = 0.009) [30]. In addition, hemoglobin levels in HD patients receiving periodontal treatment (11.7/1.5 mg/dL) were significantly higher (*p* = 0.039) when compared to untreated patients (10.9/1.6 mg/dL) [37]. Thus, anemia might be associated with periodontal disease in HD patients. In general, various factors can affect the anemic status of HD patients, including administration of recombinant erythropoietin and iron, nutrition, inflammation, and dialysis dose. Regarding iron deficiency-related markers, serum ferritin levels have been reported to be positively associated with periodontitis severity in 253 HD patients [35]. However, interpretation of this finding is difficult because serum ferritin levels are recognized to be not only a marker of bone marrow iron stores but also inflammation [54,55]. In addition, the role of inflammation and serum ferritin levels in HD patients also depends on iron sufficiency [56]. Conversely, other reports have shown that serum transferrin levels in HD patients with periodontitis (mean/SD; 211.7/68.2 mg/dL) were similar (*p* = 0.11) to those without periodontitis (263.8/81.4 mg/dL) [13]. Thus, the impact of iron metabolism and storage on anemia caused by periodontal disease in HD patients is not fully understood and we believe that further studies are necessary to reach a definitive conclusion [48].

### 3.2. Correlation with Duration of HD

Various physiological functions and pathological conditions are mediated by long term HD [56,57,58,59,60]. Regarding periodontitis, the lack of a significant correlation between severity of the disease and the duration of HD has been reported in 103 HD patients [10]. Likewise, no significant relationships between HD duration and the prevalence and/or severity of periodontal disease have been reported by other investigators [9,11,14,41,61]. Furthermore, one report indicated that periodontal indices, such as PI, GI, and PPD, in HD patients were similar to those in healthy controls, and these values showed no variation during the first 5 years of HD [16]. However, that study also demonstrated that a significant increase in these values was detected during the second 5-year period, and a significantly greater increase was observed after 10 years [16]. In short, their results showed that these periodontal indices worsened with longer duration of HD. In addition, another report showed that GI and PPD scores were significantly higher in subgroups receiving HD for 3 or more years (*p* ≤ 0.001 and < 0.001, respectively), and were positively correlated with HD duration (*r* = 0.48; *p* < 0.001 and *r* = 0.48; *p* < 0.001, respectively) [15]. Furthermore, another study confirmed the correlation between duration of HD and GI or PPD (*r* = 0.27; *p* = 0.008 or r = 0.39, *p* < 0.001, respectively) [62]. Thus, several reports have indicated a positive correlation between longer dialysis duration and periodontal disease-related parameters in HD patients [15,16,35,44], and a multivariate analysis model including age, DM, smoking status, and serum albumin level confirmed these findings [35].

Conversely, there have been conflicting results regarding the relationship between HD duration and decayed, missing, and filled teeth (DMFT). In short, although Bayrakter et al. reported that HD duration was positively correlated with missing (*r* = 0.26, *p* = 0.024), but not with decayed, filled teeth, or the DMFT index, Sekiguchi et al. found that HD duration was correlated with decayed teeth (*r* = 0.42, *p* < 0.001) and the DMFT index (*r* = 0.28, *p* = 0.006), but not with missing and filled teeth [15,62]. Thus, the relationship between the prevalence and/or severity of periodontitis and duration of HD is still under debate. This discrepancy may be due to differences in patient backgrounds and lifestyle habits, as various factors including age, diabetes, and smoking status, may have affected the pathogenesis and progression of periodontitis [35]. Furthermore, negligence of oral hygiene and a period of pre-dialysis with CKD are the main causes of higher prevalence and severity of periodontitis in HD patients rather than uremic conditions [10]. Conversely, a study has also reported that the duration of HD was not associated with specific microbiota or biofilms in 52 HD patients [63]. In this review, we would like to emphasize that duration of HD is one of the most powerful predictors of HD-induced complications and survival. Finally, more detailed investigations considering factors such as patient background, lifestyle habits, microbiota, complications, quantity, and method of HD are important to further clarify this issue.

### 3.3. Correlation with an Immune Response

A variety of immune cells, such as monocytes and dendritic cells, in periodontal tissues recognize and/or phagocytose bacteria, and subsequently secret various inflammatory mediators including ILs, TNF-α, vascular endothelial growth factor, and matrix metalloproteinase [33,64,65]. Furthermore, intervention studies have shown that treatment of periodontal disease decreased systemic levels of IL-6 and CRP [29]. Thus, the above evidence suggests that periodontal disease may mediate the immune response in patients receiving HD.

It has been suggested that a weak immune response underlies the increased incidence and rapid progression of periodontitis in patients with HD [66]. In short, the immune system in HD patients, especially those with DM-induced ESRD, might be unable to fend off the bacteria. In addition to this immune vulnerability, deteriorated dental and oral status due to greater plaque accumulation, dental calculus, salivary urea concentration, and salivary pH levels have been suggested to be related the high frequency of periodontal diseases [66,67]. Furthermore, poor oral hygiene due to gingival bleeding and decreased SFR compared to healthy persons is likely to be associated with persistent inflammation of periodontal tissues in HD patients [61,63,66,68,69]. In fact, a study with a large population (*n* = 4205 adults) showed that poorer dental health was positively associated with early all-cause death and cardiovascular diseases [70]. Finally, this chronic inflammation and consistent bacterial infection under weak immune responses is speculated to contribute to spreading of bacteria from the focal periodontal tissues into the bloodstream, and to the subsequent systematic inflammation in patients with HD.

The cytokine levels in the periodontium of HD patients have been previously investigated [53]. In that study, TNF-α and IL-8 were measured in the gingival fluids of 43 HD patients. The results showed that gingival fluid levels of TNF-α in HD patients (31.4/1.41 pg/mL) was nearly ten times as high as those in healthy controls (3.06/0.15; *p* < 0.001) [53]. A similar significant difference in IL-8 levels was also detected (90.98/94.03 and 35.05/16.86 pg/mL, respectively; *p* < 0.001) [53]. Furthermore, TNF-α levels in the gingival crevicular fluids were positively associated with PI, GI, and PPD (*p* < 0.001) in HD patients, and similar positive correlations were also detected between IL-8, and PI (*p* = 0.002), GI (*p* = 0.002), and PPD (*p* = 0.012) [53]. Based on these observations, TNF-α and IL-8 expression in the periodontal pocket is speculated to play crucial roles in the pathogenesis, development, and immune response to periodontal disease in HD patients. Nevertheless, there is no general agreement on the relationship between periodontal disease and gingival crevicular fluid levels of TNF-α and IL-8 in patients with normal renal function. In short, the levels of these cytokines in patients with periodontal disease were significantly higher than in healthy controls [71]; however, other investigators have shown that gingival crevicular fluid levels of TNF-α and IL-8 in periodontitis patients were similar to non-periodontitis patients [72,73]. Indeed, both of TNF-α and IL-8 levels in the gingival crevicular fluids of HD patients with periodontitis were reportedly not significantly different compared to patients with gingivitis (*p* = 0.213 and 0.823, respectively) [53]. However, we should also note that IL-1ra, IL-6, and interferon-γ levels in gingival crevicular fluids were significantly correlated to serum levels, whereas TNF-α and IL-8 were not identified in the serum of periodontitis patients with normal renal function, [74]; however, similar analyses have not been performed in HD patients to date. Thus, the pathological roles of TNF-α and IL-8 in periodontal disease in HD patients are not fully understood. The immune system is regulated by numerous cytokines, growth factors, and immune response-related molecules. However, unfortunately, immune function in the context of periodontal disease in HD patients is only partially understood. We strongly suggest that more detailed studies are necessary to understand the pathological characteristics of periodontal disease in patients undergoing dialysis.

### 3.4. Correlation with Cardiovascular Diseases, Metabolic Syndromes, and Pneumonia

There is a general agreement that the most important comorbidities and/or causes of death in HD patients are cardiovascular disease and DM [11,37,75]. Therefore, in this section, we reviewed the impact of periodontal disease on cardiovascular diseases in HD patients. In addition, we also reported findings regarding metabolic syndrome and pneumonia because of their frequency and significant contributions to mortality in patients with HD. DM was also discussed in the following section.

#### 3.4.1. Cardiovascular Disease

Kaplan–Meier survival analyses have shown that cardiovascular disease-free survival rates in HD patients with moderate/severe periodontitis (defined as 2 or more teeth with at least 6 mm of inter-proximal attachment loss) were significantly worse (*p* = 0.01) compared to those with no/mild periodontitis [3]. In addition, a multivariate analysis model including age, center, sex, dialysis vintage, smoking status, cause of ESRD, DM, and hypertension demonstrated that moderate/severe periodontitis was an independent predictor of cardiovascular diseases in HD patients (hazard ratio = 5.0, 95% CI; 1.2–19.1, *p* = 0.02) compared to no/mild periodontitis [3]. However, the study also showed that periodontitis does not play a significant role in all-cause mortality among patients with HD (hazard ratio = 1.8 and 95% CI; 0.7–4.5) [3]. In contrast, other investigators showed that periodontal disease was significantly associated with risks of both cardiovascular-related and all-cause mortality [11]. Briefly, Kaplan–Meier survival curves showed that the cumulative survival rates in HD patients with severe periodontitis were significantly worse (*p* < 0.001) than in HD patients with no/mild or moderate periodontitis [11]. In addition, they found that the overall mortality rates in the no/mild (*n* = 104), moderate (*n* = 98), and severe periodontitis (*n* = 51) groups were 24.0%, 41.8%, and 70.6%, respectively, a statistically significant difference (*p* < 0.001) [11]. Interestingly, this study also showed that in a multivariate analysis model including age, serum levels of albumin, hs-CRP, the Charlson Comorbidity index score, education level, and history of smoking, severe periodontitis was an independent risk factor for all-cause mortality (hazard ratio = 1.83, 95% CI; 1.04–3.24, *p* < 0.05), but not cardiovascular-related diseases (hazard ratio = 1.95, 95% CI; 0.90–4.23, *p* = 0.09) [11]. To summarize, while a prior study had demonstrated that periodontitis is closely associated with mortality from cardiovascular disease, but not with all-cause mortality, the latter study showed opposing results.

A contrasting opinion regarding how periodontitis affects the risk for all-cause and cardiovascular mortality in patients on HD has been raised [44]. In short, the risk of all-cause and cardiovascular disease morality in HD patients with moderate-severe periodontitis was lower (hazard ratio = 0.74, CI = 0.61–0.90 and 0.67, 0.51–0.88, respectively) compared to those with none/mild periodontitis [44]. Importantly, these analyses were performed in propensity-matched cohorts.

#### 3.4.2. Metabolic Syndromes and Pneumonia

Poor oral heath was reported to be associated with metabolic syndrome in 312 HD patients [76]. Briefly, HD patients with metabolic syndrome (*n* = 145) had a higher score compared to those without metabolic syndrome (*n* = 108) in terms of PI (mean/SD; 2.23/0.05 vs. 2.03/0.06; *p* = 0.01), GI (1.63/0.07 vs. 1.33/0.08; *p* = 0.003), and PDI (4.35/0.10 vs. 3.84/0.14; *p* = 0.002) [76]. In addition, they found that metabolic syndrome was positively associated with the severity of periodontitis (*p* = 0.002), and multivariate analysis also showed that moderate or severe periodontitis was an independent risk factor for metabolic syndrome in HD patients (odds ratio; 2.74, 95% CI; 1.29–5.79, *p* = 0.008) [76]. Furthermore, other investigators have compared periodontal indices in HD patients with and without metabolic syndrome [77]. The authors found that bone resorption in the metabolic syndrome group (mean/SD: 1.99/0.39 mm) was significantly higher than in the non-metabolic syndrome group (1.45/0.12 mm) [77]. In addition, PPD showed significant differences between the metabolic syndrome and non-metabolic syndrome groups (mean/SD: 2.73/0.47 and 2.17/0.18, respectively; *p* < 0.05).

Conversely, the cumulative incidence of pneumonia mortality in HD patients with periodontal disease was found to be significantly higher than in HD patients without periodontal disease (*p* < 0.01) [41]. Interestingly, another report showed contrasting findings where intensive treatment of periodontal disease led to a reduced risk of acute and sub-acute pneumonia (hazard ratio; 0.77, 95% CI; 0.65–0.78, *p* < 0.001) in patients with HD [78]. Incidentally, this study also demonstrated that periodontal disease treatment in HD patients was associated with a lower risk of endocarditis (hazard ratio; 0.54, 95% CI; 0.35–0.84, *p* < 0.01) and osteomyelitis (hazard ratio; 0.77, 95% CI; 0.62–0.96, *p* < 0.05) [78]. Thus, periodontal disease is speculated to have played an important role in the pathogenesis and mortality due to metabolic syndrome and pneumonia in HD patients.

### 3.5. Diabetic and Non-Diabetic Nephropathy

Currently, there is a general agreement that DM is a major cause of dialysis. In other words, many patients with dialysis have been receiving treatment for DM for a long time. Conversely, DM is considered to be one of the major causes of periodontal diseases [79,80]. It has been speculated that HD patients with DM have higher risk of periodontal diseases compared to those without DM. Unfortunately, few studies on the prevalence and severity of periodontal disease in HD patients with DM have been performed, and there is no systematic review on this issue to date.

#### 3.5.1. Comparison of the Prevalence

In Japan, a cross-sectional study composed of HD patients with DM (*n* = 29), HD patients with chronic kidney nephropathy (CGN; *n* = 69), and a control group (*n* = 106) was performed. The study showed that mean/SD number of teeth present in HD patients with DM (17.9/9.8) was significantly lower (*p* < 0.05) compared to those with CGN (24.1/6.8) and the control group (25.3/5.8) [81]. In addition, the same study showed that the percentage of sites with bleeding on probing in HD patients with DM (22.2%) was significantly higher (*p* < 0.05) than in those with CGN (15.9%) and in controls (9.3%) [81]. The authors indicated that the reason for this difference in periodontal conditions between HD patients with DM and those with CGN were unclear; however, they speculated that various factors, such as healthy behavior, social, economic, and environmental status, and mental and systemic conditions, might influence the oral status in these patients [81]. Furthermore, the authors found that the SFR and total score of xerostomia in HD patients with DM or those with CGN were significantly different than those of the control group; however, these periodontitis-related parameters were similar between DM and CGN patients [81]. Thus, the oral health status in patients with renal dysfunction was worse than that of individuals with normal renal function, albeit with similarities observed between HD patients with DM or GCN. Interestingly, they showed that smoking status was significantly associated with the number of teeth in HD patients with DM. Smoking is a well-known risk factor for periodontitis and teeth loss, and it is also associated with development of DM. [82,83]. Thus, smoking played an important role, both directly and indirectly, in the oral health of HD patients with DM [81].

In contrast, the prevalence of moderate or severe periodontitis in HD patients with DM (74/76 patients = 97.4%) was reported to be similar to that of those without DM (51/53 ones = 96.2%), and the severity of periodontitis was not significantly associated with DM status (*p* = 0.71) [84]. In addition, this study showed that various periodontal findings, such as the PBI, PPD, CAL, and bleeding on probing, showed no significant differences in HD patients with and without DM (*p* = 0.72, 0.40, 0.58, and 0.79, respectively) [84]. Thus, the impact of diabetes on the prevalence and/or severity of periodontal disease in HD patients must be clarified by further investigations.

Interestingly, the prevalence of a variety of bacteria differed between HD patients with and without DM (*Capnocytophaga* species *p* = 0.02; *Eubacterium nodatum*, *p* = 0.02; *Parvimonas micra*, *p* = 0.03, *Porphyromonas gingivalis, p* = 0.02) [84]. However, the authors could not definitively conclude on the relationships between periodontitis and DM in patients with HD due to several limitations; for example, the plaque index was not assessed, the mean age of patients was different, and the study was performed across multiple centers. However, they concluded that DM has no decisive influence on periodontal conditions in HD patients [84]. Nevertheless, we should consider that their study population included HD patients with well-controlled DM (mean/SD hemoglobin A1c level = 6.3/2.7).

#### 3.5.2. Objective and Subjective Manifestations

The first comparative study of oral health, dental conditions, and periodontal status in HD patients with and without DM was reported in 2005 [85]. Regarding objective manifestations, the study showed that the percentage of HD patients with DM exhibiting uremic odor (12/43 patients; 27.9%) was significantly lower (*p* = 0.018) compared to those without DM (42/85 patients; 49.4%). However, a similar significant difference was not found in tongue coating, mucosa petechia, or ecchymosis [85]. Conversely, other investigators found that all of these objective manifestations, including uremic odor, did not differ significantly between HD patients with (*n* = 46) and without DM (*n* = 54) [86]. Nevertheless, another report showed that uremic odor, tongue coating, and mucosa petechia were significantly different between the two patient groups (*p* = 0.044, 0.026, and 0.008, respectively) [87].

Another report using a visual analogue scale to assess subjective symptoms, including dry mouth (xerostomia), taste change (dysgeusia), and tongue/mucosa pain, showed that HD patients with DM had significantly higher scores than those without DM (*p* = 0.040, 0.004, and 0.018, respectively) [85]. This was in contrast to findings indicating that DM status in HD patients was significantly associated with dysgeusia (*p* = 0.008), but not with dry mouth or mucosal pain [87]. Thus, these 2 studies showed that taste change (dysgeusia) in HD patients differed significantly among patients with and without DM; however, another report showed that the frequency of dysgeusia was similar [86]. In addition, no significant relationships with regard to diabetic status or xerostomia have been reported by other studies [81,87]. Thus, there is no general agreement concerning differences in objective and subjective manifestations of HD patients with and without DM.

#### 3.5.3. The Community Periodontal Index and DMFT Index

As mentioned above, the DMFT index has been commonly used to determine the incidence of dental caries, while the community periodontal index (CPI) has been used to assess periodontal condition using a mouth mirror and a probe according to the World Health Organization criteria [88]. The DMFT index consists of 4 parameters (decayed, missing, fillings, and total teeth) and the CPI consists of 6 (healthy periodontium, bleeding on probing, calculus deposition, probing depth of 4 to 5 mm, probing depth of 6 mm or deeper, and 3 or more teeth missing).

We compared the DMFT index of HD patients with and without DM in Table 5. The overall DMFT score in HD patients, which was calculated as the sum of the number of decayed, missing, and filled teeth, was remarkably higher (*p* = 0.001) in DM patients (19.93/81.9) than in non-DM patients (14.26/9.19) [85]. This finding was supported by other investigators [87]. However, the overall DMFT score was not associated with DM status in HD patients (Table 5) [83,85]. Nevertheless, as shown in Table 4, one report showed that the DM status was significantly associated with each index item, while others showed opposing results [84,85,87].

Evaluating the CPI in HP patients, Chuang et al. [85] reported that there was a borderline significant difference in patients with diabetic and non-diabetic nephropathy (*p* = 0.055). Murali et al. [86] also found no significant association between DM status and CPI in HD patients. These results were obtained by using the chi-square test to compare diabetic nephropathy and CPI codes. In contrast, a different study reported that one CPI variable, probing depth of 6 mm or deeper, in the DM patients receiving HD group was significantly higher (*p* = 0.015) than in non-DM patients, whereas other codes, such as calculus deposition and probing depth of 4 to 5 mm, were not [87]. Thus, with the exception of one code (probing depth 6 mm or deeper), a significant difference in CPI scores was not found in previous studies. However, in contrast to this report, another study reported that the percentage of bleeding on probing and sites in HD patients with DM was significantly higher than in non-DM patients (mean/SD%: 13.3/22.2 versus 8.2/15.9; *p* < 0.05) [81]. However, we must note that this result was obtained using the Tukey HSD test, which is a multiple comparison test, to compare among patients on HD for diabetic nephropathy and chronic glomerulonephritis and healthy controls.

#### 3.5.4. Properties of Saliva

Regarding SFRs in patients with HD, unstimulated and stimulated salivary flow, and buffering capacity of stimulated saliva have been reported to have no significant relationship with diabetic status (*p* = 0.15, 0.20, and 1.0, respectively) [83], which was also supported by another study [81]. However, Sung et al. [89] reported that the unstimulated whole SFR in 68 diabetic HD patients was significantly lower (*p* = 0.032) than that in 116 non-diabetic HD patients.

The salivary pH level in DM patients (mean/SD; 7.97/0.67) showed a trend (*p* = 0.063) towards being lower than that of patients without DM (8.22/0.44) [85]. Similarly, another study showed that the frequency of salivary pH >7.0 in the DM patients was lower compared to the non-DM patients (17.0% versus 34.0%, respectively; *p* = 0.056); however, the frequency of salivary pH <7.0 in both patient groups was similar (DM; 36.2% and non-DM; 34.0%; *p* = 0.823) [87]. Chi-square analysis showed that the relationship between diabetic status and salivary pH level was not significant (*p* = 0.623) [87]. In contrast, Schmalz et al. [84] reported an opposing finding, where the mean/SD salivary pH level of HD patients with DM (6.7/0.7) was significantly lower than that of non-DM patients (7.0/0.9; *p* < 0.01) [84].

As shown in Table 6, the changes in the SFR and pH level according to diabetic status are unclear. While the precise reasons underlying this discrepancy are unknown, differences in patients’ backgrounds, history of chronic renal failure and HD, duration of DM, and hemoglobin A1c level may be factors. Conversely, an investigation into the relationships between periodontal condition-related parameters and glycemic control in HD patients with DM showed that tongue coating, dry mouth, and tongue/mucosal pain were significantly correlated with hemoglobin A1c levels (*p* = 0.001, 0.001, and 0.004, respectively); however, salivary pH levels were decreased with higher hemoglobin A1c levels (mean/SD pH level in hemoglobin A1c level ≤6%; 8.20/0.36, 6.1%–9%; 8.00/0.58, ≥9.1%; 7.46/1.07), but the differences were not statistically significant (*p* = 0.086) [85]. Therefore, we emphasized the importance of more detailed investigations with a larger study population to arrive at a conclusion.

### 3.6. Periodontal Indices and Salivary Status with Peritoneal Dialysis

In the above sections, we discussed the changes in periodontal indices and salivary findings in response to HD. However, several reports have shown differences in these parameters between PD and HD patients, and controls (Table 7). For example, Bayraktar et al. [90]. reported that plaque and calculus accumulation in the HD and PD groups were higher than in controls. However, gingival inflammation in PD patients was significantly lower than in HD patients [22,68,90]. Moreover, the SFR in the PD group was significantly higher than that in the HD group, although both groups had significantly lower values than the control group [68]. The reduction in the SFR has been known to increase the risk of caries [91]. In fact, the authors found that the number of filled teeth was higher in the PD group than in the HD group. Differences in the SFR might be associated with better volume status in the PD group and in the relatively more liberal fluid intake because of residual renal function. In fact, the fluid dynamics of the gingiva might influence gingival health in children undergoing PD therapy [92]. We have summarized these results in Table 6. These results seem to show that the pattern of differences in periodontal indices and salivary status between healthy controls and patients with PD was not replicated between HD and PD patients. However, the data was insufficient to draw definitive conclusions.

## 4. Management of Periodontal Disease

Many investigators have suggested that correct diagnosis and appropriate treatment of periodontal disease are important not only to the improvement of oral infection and inflammation, but also to maintain systemic health in patients receiving dialysis, and that managing oral health can effectively prolong survival [15,16,17,23,41,61,70]. In fact, brushing the teeth twice daily led to a reduced chance of developing periodontal diseases than brushing once daily, and it was identified as an independent factor [36]. Furthermore, a lower frequency of visits to the dentist has also been reported to be positively associated with higher mortality in HD patients [37]. Conversely, this report also showed that although the mortality of HD patients with chronic periodontitis is worse than of those without, the survival rate of patients treated for chronic periodontitis was not significantly different from that of untreated patients (*p* = 0.774) [37]. Furthermore, Hou et al. emphasized that special efforts for the prevention and management of periodontal disease are important in elderly HD patients because aging is another risk factor for periodontal disease in HD patients [36]. In addition to the prevention of different complex diseases and the prolongation of survival, maintenance of oral health and treatment of periodontal disease are essential for dialysis patients waiting for renal transplantation because patients with active inflammation and/or severe periodontal disease are usually judged as unfit for transplantation.

Thus, many studies support the importance of maintenance of dental health and periodontal treatment in patients with dialysis. However, in the real world, approximately 70% of Japanese HD facilities have no registered dental clinic [93]. Consequently, collaboration with dental facilities was promoted as beneficial for maintenance and management of oral health in HD patients [93], and found support from other investigators [53]. In addition, education on preventive dental care is important to the collaboration with dentists [94]. The prevalence and severity of periodontal disease are reported to vary substantially according to country, rather than being rotted in individual patient characteristics or healthcare [75].

Thus, conscientious efforts are necessary to establish an effective management and/or treatment approach to oral health in patients receiving dialysis treatment. In addition, this system must be modified for different countries in accordance with their specific conditions, including causes of ESRD, complications, and lifestyle habits. However, we believe that such a management approach in collaboration with dentists is well worth implementing because management of oral health and periodontal disease leads to maintenance or improvement in the QoL, reduction of complications, and prolongation of survival in dialysis patients.

## 5. Conclusions and Future Perspectives

This review showed that periodontal diseases affect the inflammation, immune response, and nutritional status in patients on dialysis. In fact, the severity of periodontal disease was significantly associated with serum levels of CRP, albumin, and a variety of minerals. In addition, several inflammation-related cytokines and molecules, such as IL-6, Il-8, TNF-α, and PTX-3 were influenced by periodontal conditions. As a result, a significant association between periodontal disease and various pathological conditions, including cardiovascular disease, metabolic syndrome, and pneumonia, is observed. Furthermore, we showed how dialysis, especially HD, exacerbates oral conditions via disruption of salivary characteristics, such as pH and flow rate. The deterioration in oral and periodontal tissues subsequently leads to a high frequency and severity of periodontal disease. Importantly, DM plays important roles in these pathological mechanisms. These findings are summarized in Figure 1. Furthermore, based on these facts, we demonstrated that maintenance of oral health and treatment of periodontal disease are important to maintaining QoL, prevention of various pathological conditions, and prolongation of survival in patients with dialysis. Unfortunately, we found that it was difficult to conduct a focused and systematic study into the relationships between periodontal disease and dialysis-related factors. Although numerous studies have been performed, they exhibit significant heterogeneity in patient characteristics, such as age, diabetic nephropathy, duration of chronic kidney disease; and methods of dialysis, such as PD or HD, the specific machine and agents used, timing of sampling, and duration. Data on the pathological roles of periodontal disease in patients with PD in particular is insufficient for a systematic review. However, continuous technological development could enable us to identify new pathogens, determine their interactions, and assess periodontal disease-related parameters in patients on dialysis. In fact, we have developed a research strategy aimed at elucidating the molecular and immune-related mechanisms of periodontal disease in patients on dialysis using novel approaches [95,96]. We emphasized that further studies with larger populations and uniform design are required to determine the pathological significance of periodontal disease and the clinical utility of oral care in patients on dialysis.

## Figures and Tables

**Figure 1 ijms-20-03805-f001:**
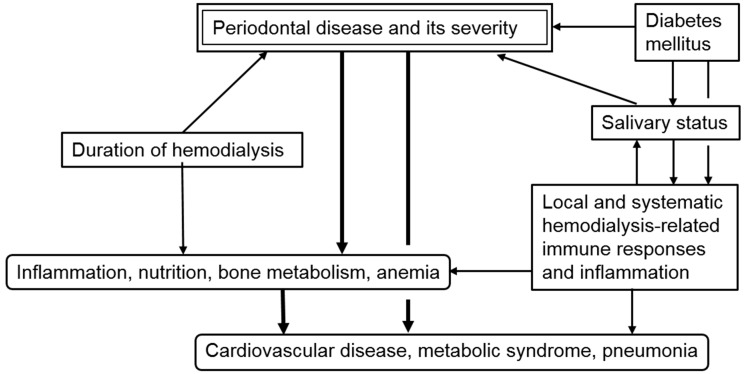
Schema of pathological roles played by periodontal disease in patients on hemodialysis.

**Table 1 ijms-20-03805-t001:** Criteria for representative periodontal measures.

**Plaque Index**	
0	No plaque in the gingival area
1	A film of plaque adhering to the free gingival margin and adjacent area of the tooth; may be recognized only by running a probe across the tooth surface
2	Moderate accumulation of soft deposits within the gingival pocket and on the gingival margin and/or adjacent tooth surface; can be seen by the naked eye
3	Abundance of soft material within the gingival pocket and/or on the gingival margin and adjacent tooth surface
**Papillary Bleeding Index**	
0	No bleeding on probing
1	Single ecchymosis of the gingiva on probing
2	Multiple ecchymoses or minor single spot extravasation from the gingiva on probing
3	Bleeding into the pocket immediately after probe insertion
4	Intensive extra pocket bleeding on probing
**Gingival Index**	
0	Normal gingiva
1	Mild inflammation, slight change in color, slight edema, no bleeding on palpation
2	Moderate inflammation, redness, edema, glazing, bleeding on palpation
3	Severe inflammation, marked redness and edema, ulceration, tendency to spontaneous bleeding
**Community Periodontal Index**	
0	Healthy gingiva
1	Bleeding observed, directly or by using mouth mirror, after probing
2	Calculus detected during probing, but all the black band on the probe visible
3	Pocket 4–5 mm (gingival margin within the black band on the probe)
4	Pocket 6 mm or more (black band on the probe not visible)
X	Excluded sextant (less than two teeth present)

**Table 2 ijms-20-03805-t002:** Relationships between serum C-reactive protein level and periodontal disease.

*n*	Correlation with Periodontal Disease and Its Severity	Author/Year/Ref
253 *	Positively correlated with periodontitis severity	Chen/2006/[35]
44	Higher in advanced periodontitis versus non-cases	Franek/2006/[32]
154	Not different between non-cases and periodontitis	Kshirsagar/2007/[33]
253 *	Positively correlated with periodontitis severity	Chen/2011/[11]
77	Not correlated with periodontitis severity	Yazdi/2013/[12]
136 *	Higher in periodontal disease versus non-cases	Hou/2017/[36]
128	Not different between healthy/gingivitis and periodontitis	Cholewa/2018/[10]
211	Higher in periodontal disease versus non-cases	Iwasaki/2018/[41]

* High-sensitive C-reactive protein; Ref: Reference.

**Table 3 ijms-20-03805-t003:** Relationships between serum albumin levels and periodontal disease.

*n*	Correlation with Periodontal Disease and Its Severity	Author/Year/Ref
253	Negatively correlated with periodontitis severity	Chen/2006/[35]
154	No difference between non-patients and periodontitis patients	Kshirsagar/2007/[33]
154	Lower in severe periodontitis versus no periodontitis	Kshirsagar/2007/[33]
253	Negatively correlated with periodontitis severity	Chen/2011/[11]
96	Lower in periodontal disease versus no periodontal disease	Rodrigues/2014/[13]
188	Not correlated with periodontitis severity	Iwasaki/2016/[14]
1355	Positively correlated with periodontitis severity	Ruospo/2017/[44]
57	Lower in periodontitis versus gingivitis cases	Naghsh/2017/[48]
128	No difference between healthy/gingivitis and periodontitis	Cholewa/2018/[10]

**Table 4 ijms-20-03805-t004:** Correlations of alkaline phosphatase, calcium, parathyroid hormone, and phosphorous with periodontal disease in patients receiving hemodialysis.

	*n*	Correlation with Periodontal Disease or Its Severity	Author/Year/Ref
ALP	96	Not different between no disease and periodontal disease	Rodrigues/2014/[13]
	128	Not different between no disease and periodontal disease	Cholewa/2018/[10]
Ca	96	Not different between no disease and periodontal disease	Rodrigues/2014/[13]
	136	Not different between no disease and periodontal disease	Hou/2017/[36]
	57	Not different between gingivitis and periodontitis	Naghsh/2017/[48]
	128	Not different between healthy/gingivitis and periodontitis	Cholewa/2018/[10]
PTH	35	Not correlated with periodontal indices	Frankenthal/2002/[51]
	253	Not correlated with periodontitis severity	Chen/2006/[35]
	136	Not different between no disease and periodontal disease	Hou/2017/[36]
	128	Not different between healthy/gingivitis and periodontitis	Cholewa/2018/[10]
P	96	Lower in periodontal disease versus no disease	Rodrigues/2014/[13]
	136	Not different between no disease and periodontal disease	Hou/2017/[36]
	57	Not different between gingivitis and periodontitis	Naghsh/2017/[48]
	128	Not different between healthy/gingivitis and periodontitis	Cholewa/2018/[10]

ALP; alkaline phosphatase, Ca; calcium, PTH; parathyroid hormone, P; phosphorous.

**Table 5 ijms-20-03805-t005:** Decayed, missing, and filled teeth (DMFT) index in hemodialysis patients with and without diabetes.

Author/Year/Ref	Decay	Missing	Filled	Overall
Chuang/2005/[85]	Not significant	↑ (*p* = 0.039)	Not significant	↑ (*p* = 0.001)
Murali/2012/[86]	–	–	–	Not significant
Swapna/2013/[87]	↑ (*p* < 0.001)	Not significant	↑ (*p* < 0.001)	↑ (*p* < 0.001)
Schmalz/2017/[84]	Not significant	Not significant	Not significant	Not significant

↑ means that variables in the diabetic group were higher compared to the non-diabetic group.

**Table 6 ijms-20-03805-t006:** Comparison of properties of saliva in diabetic and non-diabetic patients.

Properties of Saliva	No. of DM/non-DM	Compared tonon-Diabetic Patients	Author/Year/Ref
Salivary flow rate	116/68	Lower	Sung/2006/[89]
	29/69	Not significant	Teratani/2013/[81]
	66/93	Not significant	Schmalz/2017/[84]
Salivary pH level	85/43	Not significant	Chung/2005/[85]
	47/50	Not significant	Swapna/2013/[87]
	66/93	Lower	Schmalz/2017/[84]

DM; diabetic mellitus, Ref; Reference.

**Table 7 ijms-20-03805-t007:** Periodontal parameters in the peritoneal dialysis, hemodialysis, and healthy groups.

Variables	PD Compared in the Healthy Group		PD Compared to HD	
	Difference	Author/Years/Ref	Differences	Author/Years/Ref
GI	NS	Bayrakter/2008, 2009/[68,90]	Lower	Borawski/2007, 2008/[4,90]
PBI	NS	Bayrakter/2008, 2009/[68,90]	Lower	Borawski/2007/[4] Bayrakter/2008/[90]
PI	Higher	Borawski/2007/[4] Bayrakter/2008/[90]	NS	Bayrakter/2008, 2009 /[68,90]
CSI	Higher	Bayrakter/2008/[90]	NS	Bayrakter/2008/[90]
S-pH	Higher	Bayrakter/2009/[68]	Higher	Bayrakter/2009/[68]
SFR	NS	Bayrakter/2009/[68]	Higher	Bayrakter/2009/[68]
PPD	NS	Bayrakter/2008/[90]	NS	Bayrakter/2008/[90]

GI; gingival index, PBI; papillary bleeding index, PI; plaque index, CSI; calculus surface index, S-; salivary-, SFR; salivary flow rate, PPD; probing pocket depth, NS: not significant, Ref; reference.

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
