# Peer review of "Periodontal Disease in Patients Receiving Dialysis"

_ijms, 2019, doi:10.3390/ijms20153805_

Round 1

Reviewer 1 Report

The authors reviewed the periodontal status in patients receiving dialysis. It is an extensive narrative review with long listings of results of various studies. I would prefer that the authors would more summarize the results of the various studies instead of repeating details of these studies in the body of the text (I would prefer to have that data in tables, including on what criteria the severity of periodontal disease was diagnosed/scored/rated).

Why was chosen for a narrative approach and not a systematic or focussed approach as there are many papers discussed in the review? There is also no referral to how the literature was searched and what papers were considered to be eligible for this review. It is also not a total overview of all studies available in the literature (in the abstracts the authors mention that the available studies were discussed in this review). So, on what grounds were papers considered eligible to be included in this review.

There is no critical reflection of the results of the various papers as there is no referral to what periodontal indices are used in the various papers (so can the results of papers be compared) as well as no discussion concerning the timing of periodontal examination and the timing of blood sampling, saliva collection etc. Particularly in haemodialysis patients serum and saliva parameters are highly dependent on time after haemodialysis. In this respect, I miss the input from a dental expert, preferably a periodontologist, who could rate whether the appropriate approaches were applied in the reviewed papers as well as which outcomes from what papers could be compared with each other.

Author Response

Response to Reviewer 1

Thank you for evaluating our manuscript. We appreciate your positive feedback. We believe that your suggestions have helped us to improve the quality of our manuscript. In addition, to further improve the quality of this review, the entire manuscript was checked by a certified dentist who is an expert in periodontal disease at our institution (Prof. A. Yoshimura). Professor Yoshimura provided his professional opinion and suggested modifications that were made in the revised version of the manuscript.

Our point-to-point responses to your comments are provided below. Line numbers indicated in parenthesis refer to the revised manuscript. In addition to this point-by-point response, the revised version of the manuscript is attached as a separate file.

The authors reviewed the periodontal status in patients receiving dialysis. It is an extensive narrative review with long listings of results of various studies. I would prefer that the authors would more summarize the results of the various studies instead of repeating details of these studies in the body of the text (I would prefer to have that data in tables, including on what criteria the severity of periodontal disease was diagnosed/scored/rated).

Response)

   We agree with your opinion. We have added a new table (Table 1) showing the criteria for assessing periodontal disease severity, as suggested. In addition, we added an explanation of why this table was necessary to the 2.1. Impact of periodontal disease section (lines 87-89). In addition, a schematic summary of the this review was added as Figure 1 and cited in the “5. Conclusions and future perspectives” section. Moreover, in the same section, we presented the limitaions of this review and discussed future perspectives (lines 643-655). We beleve that these revisions have improved the quality of the review.

Why was chosen for a narrative approach and not a systematic or focused approach as there are many papers discussed in the review? There is also no referral to how the literature was searched and what papers were considered to be eligible for this review. It is also not a total overview of all studies available in the literature (in the abstracts the authors mention that the available studies were discussed in this review). So, on what grounds were papers considered eligible to be included in this review.

Response)

Thank you for your important comments. First, we must explain the difficulty of a systematic or focused approach in this field. In short, the data on the pathological significance of periodontal disease in patients on dialysis, especially PD, is too scarce to conduct a proper systematic review. In fact, when we searched PubMed for studies on “periodontal disease” and “peritoneal dialysis”, only 26 were retrieved. We subsequently narrowed down the studies by excluding those with no clinical or laboratory data. In addition, in accordance with the editorial policy of the International Journal of Molecular Sciences, we have tried to emphasize the most recent studies by as much as possible. In addition, the studies showed significant heterogeneity in patient characteristics and dialytic methods, including age, original diseases (diabetic or non-diabetic), duration of chronic kidney disease, the machine and agents used, and duration of hemodialysis.

   Based on these issues and your suggestion, we have clarified our approach to searching the literature and the determining the eligibility of included studies in the revised “1. Introduction” section (lines 75-78). In addition, the heterogeneity in patient backgrounds and dialysis methods was described as a major limitation of this review in the “5. Conclusions and future perspectivessection, which was added in accordance with your suggestion (lines 643-649).

In recent years, the importance of the systemic changes observed during HD, including inflammation, nutrition, and bone metabolism, has been pointed out. Therefore, we reviewed the data on serum levels of CRP, albumin, calcium, phosphorus, and parathyroid hormone, and hematological data in the “3.1. Correlations with blood test indices” section. In this revison, we added such comment into “3.1. Correlations with blood test indices” section (lines 152-154)

There is no critical reflection of the results of the various papers as there is no referral to what periodontal indices are used in the various papers (so can the results of papers be compared) as well as no discussion concerning the timing of periodontal examination and the timing of blood sampling, saliva collection etc. Particularly in haemodialysis patients serum and saliva parameters are highly dependent on time after haemodialysis. In this respect, I miss the input from a dental expert, preferably a periodontologist, who could rate whether the appropriate approaches were applied in the reviewed papers as well as which outcomes from what papers could be compared with each other.

(Response)

   We agree with your opinion. We have added a discussion of blood and saliva sampling from patients on PD to the “2.1. Impact of periodontal disease section (lines 99-101 and lines 109-110). In HD patients, routine blood sampling is generally performed at the start of HD. However, if it is performed at the time of a dental consultation, blood sampling at HD initiation is not possible. In the real-world, the management of systemic diseases and control of HD conditions is performed by nephrologists or urologists. Therefore, blood sampling is presumed to have been performed at the start of HD. We have added these comments to the 3.1. Correlations with blood test indices” section (lines 156-161). In addition, since this issue was a limitation of our review, we have also mentioned them in the “5. Conclusions and future perspectives” section (lines 643-648).

Moreover, to improve the quality of this review, the entire manuscript was reviewed by a certified dentist who is an expert in periodontal disease at our institution (Prof. A. Yoshimura). He gave his professional opinion and suggested modifications to the text’s language. Finally, Professor Yoshimura approved that the manuscript’s contents were correct and that our review would be useful to not only nephrologists, urologists, and basic scientists, but also dentists seeking to understand the role played by periodontal disease in patients on dialysis.

Reviewer 2 Report

Thank you for the important decision in prepare a review on such an important subject for us all working in the field of dialysis and nephrology.

The information you have gathered is very useful and informative for people interested in this field. Howeve, I think that there is a need for a major revision of the English language, trying to adjust the vocabulary to the area :

For example  in lines 42 to 44 in the introduction section, you must change the word "drain" for " remove", "mineral" for electrolytes". What you mean, for example, in line 43 with "to introduce insufficient matters into the body" ?

I am sure that the paper being written in a more comprehensive English , it will enable the readers (like me in the review) to understand better all the information that you are willing to convey.

I would also like to have all the correlations that you refer to in different sections of the paper communicated in a summarized way and with your own comments of what you think these correlations mean. Sometimes, I feel that you focused on transcripting some results and letting the readers draw their own conclusions.

More than a report of a group of publications, your review could be a source of information and an inspiring paper to new ideas and questions been brought forward by the readers of this review.

Author Response

Response to reviewer 2

Thank you for evaluating our manuscript. We appreciate your positive feedback. We believe that your suggestions have helped us to improve the quality of our manuscript. In addition, to further improve the quality of this review, the entire manuscript was checked by a certified dentist who is an expert in periodontal disease at our institution (Prof. A. Yoshimura). Professor Yoshimura provided his professional opinion and suggested modifications that were made in the revised version of the manuscript.

Our point-to-point responses to your comments are provided below. Line numbers indicated in parenthesis refer to the revised manuscript. In addition to this point-by-point response, the revised version of the manuscript is attached as a separate file.

The information you have gathered is very useful and informative for people interested in this field. However, I think that there is a need for a major revision of the English language, trying to adjust the vocabulary to the area:

For example, in lines 42 to 44 in the introduction section, you must change the word "drain" for " remove", "mineral" for electrolytes". What you mean, for example, in line 43 with "to introduce insufficient matters into the body"?

Response)

   Thank you very much for your important suggestions. We wanted to explain that “peritoneal dialysis (PD) and hemodialysis (HD) are performed to supply insufficient nutrients to the body”. We have modified the sentence based on your suggestions (“1. Introduction” section; lines 45-46). In addition, we checked the entire text once more, including the tables and abstract, and made the necessary revisions.

I am sure that the paper being written in a more comprehensive English, it will enable the readers (like me in the review) to understand better all the information that you are willing to convey.

(Response)

   We agree with your opinion. We have revised the entire text to ensure that it is clear, coherent, and readily understandable to readers. For examples, we modified our use of decimal points for greater clarity (“3.1.2. Serum albumin level”; lines 217-220, “3.2. Correlation with duration of dialysis”; lines 331-333, “3.4.2. Metabolic syndrome and pneumonia”; line 441). In addition, our manuscript’s language has been reviewed once more by a native English speaker provided by a professional editing firm (Editage; www.editage.jp).

Furthermore, we believe that adding the new Table 1, which outlines the periodontal parameters mentioned in the text, will improve the clarity of our review.

I would also like to have all the correlations that you refer to in different sections of the paper communicated in a summarized way and with your own comments of what you think these correlations mean. Sometimes, I feel that you focused on transcripting some results and letting the readers draw their own conclusions.

(Response)

   Thank you for your valuable comments. According to your suggestion, we have provided an illustration showing the pathological roles played by periodontal disease in hemodialysis patients (Figure 1, cited in the “5. Conclusions and future perspectives” section).

   In addition, we added our own views on various previous reports in the revised version of the manuscript. For examples;

# “2.2. Impact of periodontal care and treatment” section: lines 133-137.

   # “3.2. Correlation with duration of hemodialysis” section: lines 349-350.

   # “3.3. Correlation with immune response” section: lines 394-39

   # “3.6. Periodontal indices and salivary status with peritoneal dialysis section: lines 389-392.

 In addition, we have added a discussion of the study’s limitations and future perspectives in the revised version of the manuscript (“5.3. Conclusions and future perspectives”: lines 643-655)

More than a report of a group of publications, your review could be a source of information and an inspiring paper to new ideas and questions been brought forward by the readers of this review.

(Response)

   Thank you for your constructive suggestion, with which we strongly agree. Therefore, we added a new section to the revised manuscript (“5. Conclusions and future perspectives”) to present our views on future research directions focused on periodontal disease in dialysis patients (Lines 749-655). In addition to summarizing the review’s findings in this section, we emphasize the importance of further, more detailed studies. In addition, we showed our opinion to lead to new ideas for readers of this review in “5. Conclusions and future perspectives” section (lines 651-653).

Round 2

Reviewer 1 Report

The authors well addressed of my concerns.

Author Response

Thank you for evaluating our manuscript. We appreciate your positive feedback. We believe that your suggestions have helped us to improve the quality of our manuscript.

Our point-to-point responses to your comments are provided below. Line numbers indicated in parenthesis refer to the revised manuscript. In addition to this point-by-point response, the revised version of the manuscript is attached as a separate file.

Comment) The authors well addressed of my concerns.

Response)

We thank you for your positive evaluation of our original revision. With regards to your comment on improvements of “English language and style”, you suggested that “Moderate English changes required” . Therefore, we have revised the entire manuscript, and have had our manuscript edited once more by a professional editing service.

Reviewer 2 Report

This sentence is " Dialytic therapy, mainly peritoneal dialysis (PD) and hemodialysis (HD), are common renal 44 replacement therapies that are used worldwide. Although both are performed to remove excessive fluids, electrolytes, and uremic toxins, and to supply insufficient nutrients to the body, the 46 mechanisms involved in the two procedures differ. "  still unclear and I would suggest to remove the last part that does not make sense. " and to supply insufficient nutrients to the body, the 46 mechanisms involved in the two procedures differ. "

Please do not use the word tube but the word catheter that is the only correct word to be used . Here is the sentence: " However, PD is less efficient at removing wastes from the body than HD, and the presence of the tube catheter presents a risk of peritonitis due to the possibility of microbial entry into the abdomen ". 

Surprisingly, a recent report suggested that 106 of 68 107 HD patients (99.1%) exhibit some form of periodontitis [9], and another recent study also showed that only 1 one of 103 HD patients evaluated had a healthy periodontium [10]

Author Response

Thank you for evaluating our manuscript. We appreciate your positive feedback. We believe that your suggestions have helped us to improve the quality of our manuscript. In addition, to further improve the quality of this review, our manuscript has been edited again by a professional editing service.

Our point-to-point responses to your comments are provided below. Line numbers indicated in parenthesis refer to the revised manuscript. In addition to this point-by-point response, the revised version of the manuscript is attached as a separate file.

Comment) This sentence is " Dialytic therapy, mainly peritoneal dialysis (PD) and hemodialysis (HD), are common renal 44 replacement therapies that are used worldwide. Although both are performed to remove excessive fluids, electrolytes, and uremic toxins, and to supply insufficient nutrients to the body, the 46 mechanisms involved in the two procedures differ. " still unclear and I would suggest to remove the last part that does not make sense. " and to supply insufficient nutrients to the body, the 46 mechanisms involved in the two procedures differ. "

Response)

Thank you for your suggestion. Based on your suggestion, we have deleted the phrase “, and uremic toxins, and to supply insufficient nutrients to the body, the mechanisms involved in the two procedures differ” in the revised version of the manuscript (1. Introduction; line 46).

Comment) Please do not use the word tube but the word catheter that is the only correct word to be used . Here is the sentence: " However, PD is less efficient at removing wastes from the body than HD, and the presence of the tube catheter presents a risk of peritonitis due to the possibility of microbial entry into the abdomen ". 

Response)

We agree with your suggestion. We have modified the sentence (1. Introduction; line 54).

Comment) Surprisingly, a recent report suggested that 106 of 68 107 HD patients (99.1%) exhibit some form of periodontitis [9], and another recent study also showed that only 1 one of 103 HD patients evaluated had a healthy periodontium [10]. 

Response)

We apologize for the simple errors. We have modified them in the revised version of the manuscript (1. Introduction; lines 68 – 69).

In the revised manuscript, we have checked all sentences, and they have been edited once more by a professional editing service.